# LEARNING TO COMPUTE WORD EMBEDDINGS ON THE FLY

## ABSTRACT

Words in natural language follow a Zipfian distribution whereby some words are frequent but most are rare. Learning representations for words in the "long tail" of this distribution requires enormous amounts of data. Representations of rare words trained directly on end tasks are usually poor, requiring us to pre-train embeddings on external data, or treat all rare words as out-of-vocabulary words with a unique representation. We provide a method for predicting embeddings of rare words on the fly from small amounts of auxiliary data with a network trained end-to-end for the downstream task. We show that this improves results against baselines where embeddings are trained on the end task for reading comprehension, recognizing textual entailment and language modeling.

## 1 INTRODUCTION

Natural language yields a Zipfian distribution (Zipf, 1949) which tells us that a core set of words (at the head of the distribution) are frequent and ubiquitous, while a significantly larger number (in the long tail) are rare. Learning representations for rare words is a well-known challenge of natural language understanding, since the standard end-to-end supervised learning methods require many occurrences of each word to generalize well.

The typical remedy to the rare word problem is to learn embeddings for some proportion of the head of the distribution, possibly shifted towards the domain-specific vocabulary of the dataset or task at hand, and to treat all other words as out-of-vocabulary (OOV), replacing them with an unknown word "UNK" token with a shared embedding. This essentially heuristic solution is inelegant, as words from technical domains, names of people, places, institutions, and so on will lack a specific representation unless sufficient data are available to justify their inclusion in the vocabulary. This forces model designers to rely on overly large vocabularies, as observed by (Mi et al., 2016; Sennrich et al., 2015), which are parametrically expensive, or to employ vocabulary selection strategies (L'Hostis et al., 2016). In both cases, we face the issue that words in the tail of the Zipfian distribution will typically still be too rare to learn good representations for through standard embedding methods. Some models, such as in the work of Ling et al. (2015), have sought to deal with the open vocabulary problem by obtaining representations of words from characters. This is successful at capturing the semantics of morphological derivations (e.g. "running" from "run") but puts significant pressure on the encoder to capture semantic distinctions amongst syntactically similar but semantically unrelated words (e.g. "run" vs. "rung"). Additionally, nothing about the spelling of named entities, e.g. "The Beatles", tells you anything about their semantics (namely that they are a rock band).

In this paper we propose a new method for computing embeddings "on the fly", which jointly addresses the large vocabulary problem and the paucity of data for learning representations in the long tail of the Zipfian distribution. This method, which we illustrate in Figure 1, can be summarized as follows: instead of directly learning separate representations for all words in a potentially unbounded vocabulary, we train a network to predict the representations of words based on auxiliary data. Such auxiliary data need only satisfy the general requirement that it describe some aspect of the semantics of the word for which a representation is needed. Examples of such data could be dictionary definitions, Wikipedia infoboxes, linguistic descriptions of named entities obtained from Wikipedia articles, or something as simple as the spelling of a word. We will refer to the content of auxiliary data as "definitions" throughout the paper, regardless of the source. Several sources of auxiliary data can be used simultaneously as input to a neural network that will compute a combined representation.

These representations can then be used for out-of-vocabulary words, or combined with within-vocabulary word embeddings directly trained on the task of interest or pretrained from an external data source (Mikolov et al., 2013; Pennington et al., 2014). Crucially, the auxiliary data encoders are trained jointly with the objective, ensuring the preservation of semantic alignment with representations of within-vocabulary words. In the present paper, we will focus on a subset of these approaches and auxiliary data sources, restricting ourselves to producing out-of-vocabulary words embeddings from dictionary data, spelling, or both.

The obvious use case for our method would be datasets and tasks where there are many rare terms such as technical writing or bio/medical text (Deléger et al., 2016). On such datasets, attempting to learn global vectors—for example GloVe embeddings (Pennington et al., 2014)—from external data, would only provide coverage for common words and would be unlikely to be exposed to sufficient (or any) examples of domain-specific technical terms to learn good enough representations. However, there are no (or significantly fewer) established neural network-based baselines on these tasks, which makes it harder to validate baseline results. Instead, we present results on a trio of well-established tasks, namely reading comprehension, recognizing textual entailment, and a variant on language modelling. For each task, we compare baseline models with embeddings trained directly only on the task objective to those same models with our on the fly embedding method. Additionally, we report results for the same models with pretrained GLoVe vectors as input which we do not update. We aim to show how the gap in results between the baseline and the data-rich GLoVe-based models can be partially but substantially closed merely through the introduction of relatively small amounts of auxiliary definitions. Quantitative results show that auxiliary data improves performance. Qualitative evaluation indicates our method allows models to draw and exploit connections defined in auxiliary data, along the lines of synonymy and semantic relatedness.

## 2   RELATED WORK

Arguably, the most popular approach for representing rare words is by using word embeddings trained on very large corpora of raw text. (Mikolov et al., 2013; Pennington et al., 2014). Such embeddings are typically explicitly or implicitly based on word co-occurence statistics. Being a big step forward from the models that are trained from scratch only on the task at hand, the approach can be criticized for being extremely data-hungry[1]. Obtaining the necessary amounts of data may be difficult, e.g. in technical domains. Besides, auxiliary training criteria used in the pretraining approaches are not guaranteed to yield representations that are useful for the task at hand.

(Dhingra et al., 2017) proposed to represent OOV words by fixed random vectors. While this has shown to be effective for machine comprehension, this method does not account for word semantics at all, and therefore, does not cover the same ground as the method that we propose.

There have been a number of attempts to achieve out-of-vocabulary generalization by relying on the spelling. (Ling et al., 2015) used a bidirectional LSTM to read the spelling of rare words and showed that this can be helpful for language modeling and POS tagging. We too will investigate spelling as a source of auxiliary data. In this respect, the approach presented here subsumes theirs, and can be seen as a generalization to other types of definitions.

The closest to our work is the study by Hill et al. (2016), in which a network is trained to produce an embedding of a dictionary definition that is close to the embedding of the headword. The network is shown to be an effective reverse dictionary and a crossword solver. Our approach is different in that we train a dictionary reader in an end-to-end fashion for a specific task, side-stepping the potentially suboptimal auxiliary ranking cost that was used in that earlier work. Their method also relies on the availability of high-quality pretrained embeddings which might not always be feasible. Another related work by Long et al. (2016) uses dictionary definitions to provide initialization to a database embedding method, which is different from directly learning to use the definitions like we do. Concurrently with this work Weissenborn et al. (2017) studied dynamic integration of background knowledge from a commonsense knowledge base. In another concurrent work Long et al. (2017) build a new dataset for named entity prediction and show that external knowledge can be very useful for this task.

---

[1]The GLoVe embeddings used are computed using 840 billion words of English text.

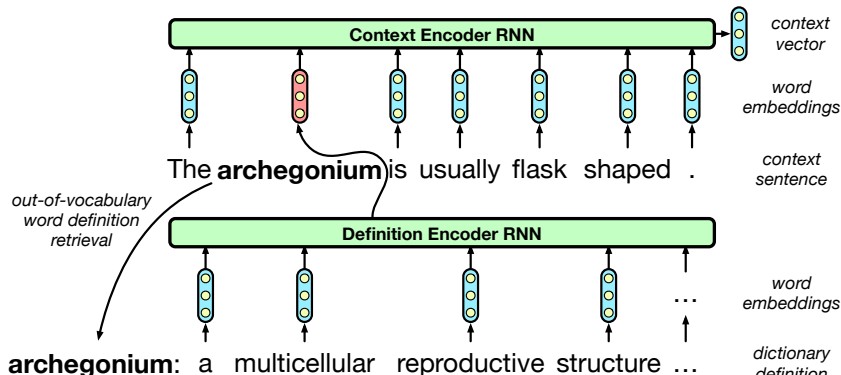

Figure 1: An example of how we deal with out-of-vocabulary words (indicated in bold). We obtain representations by retrieving external information (e.g. a dictionary definition) and embedding it, for example, with another LSTM-RNN, instead of using a catch-all "UNK" representation for out-of-vocabulary items.

At a higher level our approach belongs to the a broad family of methods for conditioning neural networks on external knowledge. For example, Larochelle et al. (2008) propose to add classes to a classifier by representing them using their "descriptions". By description they meant, for example, a canonical picture of a printed character, that would represent all its possible handwritten versions. Their idea to rely on descriptions is similar to our idea to rely on definitions, however we focus on understanding complex inputs instead of adding new output classes.

Enhancing word embeddings with auxiliary data from knowledge bases (including wordnet) has a long tradition (Xu et al., 2014; Faruqui et al., 2015). Our work differs from previous approaches in essential ways. First, we use a textual form and are not restricted to knowledge represented as a graph. Second, we learn in an end to end fashion, allowing the model to pick useful information for the task of interest.

## 3 ON THE FLY EMBEDDINGS

In general, a neural network processes a language input by replacing its elements $x_i$, most often words, with the respective vectors $e(x_i)$, often called embeddings (Bengio et al., 2003). Embeddings are typically either trained from scratch or pretrained. When embeddings are trained from scratch, a restricted vocabulary $V_{train} = \{w_1, \ldots, w_n\}$ is defined, usually based on training set frequency. Words not in $V_{train}$ are replaced by a special token $UNK$ with a trainable embedding $e(UNK)$. Unseen test-time words $w \notin V_{train}$ are then represented by $e(UNK)$, which effectively means the specific meaning of this word is lost. Even if $w$ had been included in $V_{train}$ but was very rare, its learned embedding $e(w)$ would likely not be very informative.

The approach proposed in this work, described in Figure 1, is to use definitions from auxiliary data, such as dictionaries, to compute embeddings of rare words *on the fly*, as opposed to having a persistent embedding for each of them. More specifically, this involves fetching a definition $d(w) = (x'_1, \ldots, x'_k)$ and feeding it into a network $f$ that produces an embedding $e_d(w) = f(e'(x'_1), \ldots, e'(x'_k))$. We will refer to $e_d(w)$ as a *definition embedding* produced by a *definition reader* $f$. One can either use the same embeddings $e' = e$ when reading the dictionary or train different ones. Likewise, one can either stick to a shared vocabulary $V_{dict} = V_{train}$, or consider two different ones. When a word $x'_i \notin V_{dict}$ is encountered, it is replaced by $UNK$ and the respective trainable embedding $e'(UNK)$ is used. For the function $f$ we consider three choices: a simple mean pooling (MP) $e_d(w) = \sum_{i=1}^{k} e'(x_i)/k$, a mean pooling (MP-L) with a linear transformation $e_d(w) = \sum_{i=1}^{k} V e'(x_i)/k$, where $V$ is a trainable matrix, and lastly, using the last state of an LSTM (Hochreiter & Schmidhuber, 1997), $e_d(w) = LSTM(e'(x'_1), \ldots, e'(x'_k))$. Many words have multiple dictionary definitions. We combine embeddings for multiple definitions using mean pooling.

We include all definitions whose headwords match $w$ or any possible lemma of a lower-cased $w^2$. To simplify the notation, the rest of the paper assumes that there is only one definition for each word.

While the primary purpose of definition embeddings $e_d(w)$ is to inform the network about the rare words, they might also contain useful information for the words in $V_{train}$. When we use both, we combine the information coming from the embeddings $e(w)$ and the definition embeddings $e_d(w)$ by computing $e_c(w) = e(w) + We_d(w)$, where $W$ is a trainable matrix, or just by simply summing the two, $e_c(w) = e(w) + e_d(w)$. Alternatively it is possible to just use $e_c(w) = e(w)$ for $w$ from $V_{train}$ and $e_c(w) = e_d(w)$ otherwise. When no definition is available for a word $w$, we posit that $e_d(w)$ is a zero vector. A crucial observation that makes an implementation of the proposed approach feasible is that the definitions $d(x_i)$ of all words $x_i$ from the input can be processed in parallel[3].

To keep things simple, we limited the scope and complexity of this first incarnation of our approach as follows. First, we do not consider definitions of word combinations, such as phrasal verbs like "give up" and geographical entities like "San Francisco". Second, our *definition reader* could itself better handle the unknown words $w \notin V_{dict}$ by using their definition embeddings $e_d(w)$ instead $e'(UNK)$, thereby implementing a form of recursion. We will investigate both in our future work.

## 4 EXPERIMENTS

We worked on extractive question answering, semantic entailment classification and language modelling. For each task, we picked a baseline model and a architecture from the literature which we knew would provide sensible results, to explore how augmenting it with on the fly embeddings would affect performance. We explored two complementary sources of auxiliary data. First, we used word definitions from WordNet (Miller, 1995). While WordNet is mostly known for its structured information about synonyms, it does contain natural language definitions for all its 147306 lemmas (this also includes multi-word headwords which we do not consider in this work)[4]. Second, we experimented with the character-level spelling of words as auxiliary data. To this end, in order to fit in with our use of dictionaries, we added fake definitions of the form "Word" → "W", "o", "r", "d".

In order to measure the performance of models in "data-rich" scenarios where a large amount of unlabelled language data is available for the training of word representations, we used as pretrained word embeddings 300-dimensional GLoVe vectors trained on 840 billion words (Pennington et al., 2014). We compared our auxiliary data-augmented on the fly embedding technique to baselines and models with fixed GLoVe embeddings to measure how well our technique closes the gap between a data-poor and data-rich scenario.

### 4.1 QUESTION ANSWERING

We used the Stanford Question Answering Dataset (SQuAD) (Rajpurkar et al., 2016) that consists of approximately 100000 human-generated question-answer pairs. For each pair, a paragraph from Wikipedia is provided that contains the answer as a continuous span of words.

Our basic model is a simplified version of a coattention network proposed in (Xiong et al., 2016). First, we represent the context of length $n$ and the question of length $m$ as matrices $C \in \mathbb{R}^{n,d}$ and $Q \in \mathbb{R}^{m,d}$ by running them through an LSTM and a linear transform. Next, we compute the affinity scores $L = CQ^T \in \mathbb{R}^{n,m}$. By normalizing $L$ with row-wise and column-wise softmaxes we construct context-to-question and question-to-context attention maps $A_C$ and $A_Q$. These are used to construct a joint question-document representation $U_0$ as a concatenation along the feature axis of the matrices $C$, $A_C Q$ and $A_C A_Q^T C$. We transform $U_0$ with a bidirectional LSTM and another ReLU(Glorot et al., 2011) layer to obtain the final context-document representation $U_2$. Finally, two linear layers followed by a softmax assign to each position of the document probabilities of it being the beginning and the end of the answer span. We refer the reader to the work of Xiong et al. (2016)

---

[2]We used the wordnet-based lemmatizer from NLTK.

[3]The same applies for all the words from a mini-batch of examples. By composing huge batches of up 10000 definitions from a mini-batch of examples, we were able to process them all in a reasonable time on GPUs.

[4]Advantages of using WordNet include its free availability and the ease of parsing, e.g., we used the NLTK (Bird, 2006) interface to extract the definitions.

for more details. Compared to their model, the two main simplifications that we applied is skipping the iterative inference procedure and using usual ReLU instead of the highway-maxout units.

Our baseline is a model with the embeddings trained purely from scratch. We found that it performs best with a small vocabulary of 10k most common words from the training set. This can be explained by a rather moderate size of the dataset: all the models we tried tended to overfit severely. In our preliminary experiments, we found that a smaller $V_{train}$ of 3k words is better for the models using auxiliary data, and that combining the information from definitions and word embeddings is helpful. Unless otherwise specified, we use summation preceded by a linear transformation for composing word embedding with definition embeddings: $e_c(w) = e(w) + We_d(w)$.

We tried different models (MP and LSTM) for reading the dictionary and used an LSTM for reading the spelling. In addition to using either the spelling or the dictionary definitions, we tried mixing the dictionary definitions with the spelling. When both dictionary and spelling were used, we found that using a LSTM for reading the spelling and MP-L for reading dictionary definitions works best. As mentioned in Section 3 our dictionary lookup procedure involves lowercasing and lemmatization. In order to estimate the contribution of these steps we add to the comparison a model that fetches the spelling of a lemmatized and lowercased version of each word. The last model in our comparison is trained with the GLoVe embeddings. Except for GLoVe embeddings, all vectors, such as word embeddings and LSTM states, have d=200 dimensions in all models.

The results are reported in Table 1. We report the exact match ratio as computed by the evaluation tools provided with the dataset, which is basically the accuracy of selecting the right answer. We report the average over multiple runs on the development set. Looking at the results one can see that adding any external information results in a significant improvement over the baseline model (B) (3.7 - 10.5 points). When the dictionary alone is used, mean pooling (D3) performs similarly to LSTM (D4). For the model with mean pooling, we verified that the matrix $W$ in computing $e_c(w)$ is necessary for best results (see model D2), and back-propagating through the process of reading definition is helpful (see model D1). We thereby establish that our method is preferable to the one by Long et al. (2016), which prescribes mean pooling of available embeddings without end-to-end training.

We found that adding the spelling (S) helps more than adding a dictionary (D) (3 points difference), possibly due to relatively lower coverage of our dictionary. However, the model that uses both (SD) has a 1.1 point advantage over the model that uses just the spelling (S), demonstrating that combining several forms of auxiliary data allows the model to exploit the complementary information they provide. The model with GLoVe embeddings (G) is still ahead with a 1.1 point margin, but the gap has been shrunk.

Finally, we evaluated the models S, SL and SD on the test set. The parameters for test evaluation were selected based on the development set results[5]. The test set results confirm the benefit of using dictionary definitions (SD has a 1 point advantage over S). Lastly, the model that uses lemmatization and lowercasing (SL) to retrieve spelling does not outperform S, which shows that the advantage of SD over S is not due to the normalization procedures and that SD must be using the dictionary definitions in a non-trivial way.

To understand how our model successfully uses the dictionary and what prevents it from using it better, we conducted a qualitative investigation on selected examples. Namely, we considered examples on which SD selected the correct answer span and S did not, and from them we chose the ones with the largest difference of log-likelihoods that S and SD assigned to the correct answer. Figure 2 shows the attention maps $A_C$ for both models for one of the examples. We can see that S has no clue that "overseas" may be an answer to an question about location, whereas SD is aware of that, presumably thanks to the definition "overseas -> in a foreign country". In a similar manner we observed SD being able to match "direction" and "eastwards", "where" and "outdoors". Another pattern that we saw is that the model with the dictionary was able to answer questions of the form "which scientist" or "which actress" better. Both words "scientist" and "actress" were not frequent enough to make it to $V_{train}$, but the definitions "actress -> a female actor" "scientist -> a person with advanced knowledge of one or more sciences" apparently provided enough information about these words that the model could start matching them with named entities in the passage.

---

[5]We were limited in how many models could be evaluated on the test set by the fact that test set evaluation is done by SQuAD authors manually.

Table 1: Exact match (EM) ratio for different models on SQuAD development and tests set. "dict" stands for dictionary, "MP" stands for mean pooling.

| model | EM dev | EM test |
|---|---|---|
| baseline (B) | 52.58 | - |
| dict, MP, sum, no back-prop (D1) | 56.27 | - |
| dict, MP, sum (D2) | 57.03 | - |
| dict, MP, transform and sum (D3) | 58.9 | - |
| dict, LSTM (D4) | 58.78 | - |
| spelling (S) | 61.94 | 62.9 |
| spelling+lemmas (SL) | 62.4 | 62.6 |
| spelling+dict (SD) | **63.06** | **64.08** |
| GloVe (G) | 64.19 | - |

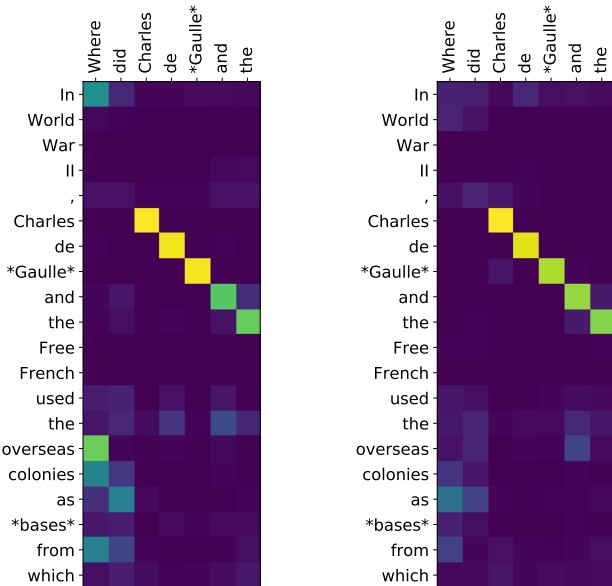

Figure 2: The attention maps $A_C$ of the models with (on the left) and without the dictionary (on the right). The rows correspond to words of the context and the columns to the words of the question. One can see how with the help of the dictionary the model starts considering "overseas" as a candidate answer to "where".

Furthermore, we compared the models G and SD in a similar way. We found that often SD simply was missing a definition. For a example, it was not able to match "XBox" and "console", "Mark Twain" and "author", "most-watched" and "most watched". We also saw cases where the definition was available but was not used, seemingly because the key word in the definition was outside $V_{train} = V_{dict}$. For example, "arrow" was defined as "a projectile with a straight thin shaft", and the word "projectile" was quite rare in the training corpus. As a consequence, the model had no chances to understand that an arrow is a weapon and match the word "arrow" in the context with the word "weapon" in the question. Likewise, "Earth" was defined as "planet", but "planet" was outside $V_{train}$. Finally, we saw cases where inferring important aspects of meaning from the dictionary would be non-trivial, for example, guessing that "historian" is a "profession" from the definition "a person who is an authority on history and who studies it and writes about it" would involve serious common sense reasoning.

Table 2: Results on SNLI and MultiNLI. X/Y means X percent accuracy on the development set and Y percent accuracy on the test set. "dict" stands for "dictionary".

|  | SNLI | MultiNLI | |
|  |  | matched | mismatched |
| --- | --- | --- | --- |
| baseline | 83.39/82.84 | 69.05/68.55 | 67.22/68.57 |
| spelling | 83.78/82.89 | 69.76/68.89 | 70.48/69.76 |
| dict | **84.88/84.39** | **71.39/71.45** | **71.65/70.7** |
| GloVe | 87.20/86.39 | 74.63/74.58 | 73.32/73.92 |

## 4.2 ENTAILMENT PREDICTION

We used the Stanford Natural Language Inference (SNLI) corpus (Bowman et al., 2015), which consists of around 500k pairs of sentences (hypothesis and premise) and the task is to predict the logical relation (contradiction, neutral or entailment) between them. In addition we used the Multi-Genre Natural Language Inference (MultiNLI) corpus (Williams et al., 2017), which effectively is a more recent and more diverse version of SNLI of approximately the same size. A key distinction of MultiNLI from SNLI is the availability of a "matched" and a "mismatched" development and test set. The matched test and development sets contain data from the same domains as the training set, whereas the mismatched ones were intentionally collected using data from different domains.

We implemented a variant (replacing TreeLSTM by biLSTM) of Enhanced Sequential Inference Model (ESIM) (Chen et al., 2016) that achieves close to SOTA accuracy. Similarly to the model used in the SQuAD experiments, this model represents hypothesis and premise as $H \in \mathbb{R}^{n,d}$ and $P \in \mathbb{R}^{m,d}$ matrices by encoding them using a bidirectional LSTM. Analogously, alignment matrices $A_H$ and $A_P$ are computed by normalizing affinity scores. These alignment matrices are used to form joint hypothesis-premise representations. For the hypothesis we compute and concatenate $H$, $A_H P$, $H - A_H P$ and $H \odot A_H P$, yielding a $\mathbf{h} \in \mathbb{R}^{n,4d}$ sentence embedding, and proceed similarly for the premise. The resulting sentence representations are then processed in parallel by a bidirectional LSTM and the final states of the LSTMs are concatenated and processed by a single layer Tanh MLP to predict entailment.

Similarly to the SQuAD experiments, we found that the baseline model performs best with a larger vocabulary (5k words for SNLI, 20k words for MultiNLI) than the model that uses auxiliary information (3k words). Differently from SQuAD, we found it helpful to use a different vocabulary $V_{dict} \neq V_{train}$. We built $V_{dict}$ by collecting the 11000 words that occur most often in the definitions, where each definition is weighted with the frequency of its headword in the training data. While in theory it would still be possible to share word embeddings between the main model and the definition reader even when they have different vocabularies, we opted for the simpler option of using separate word embeddings $e$ and $e'$. Since with separate word embeddings having a subsequent linear transformations would be redundant, we simply add the result of mean pooling $\sum_{i=1}^{k} e'(x_i)/k$ to $e(w)$. We use an LSTM for reading the spelling, but unlike the SQuAD experiments, we found that simply adding the last hidden state of the spelling LSTM to $e(w)$ worked better. We also tried to use a LSTM for reading definitions but it did no better than the simpler mean pooling. Our last model used pretrained GloVe embeddings. 300-dimensional ESIM performed best for the baseline and GloVe models, whereas using just 100 dimensions worked better for the models using auxiliary information. All runs were repeated 3 times and scores are averaged.

Results on SNLI and MultiNLI are presented in Table 2. Using dictionary definitions allows us to bridge ≈40% of the gap between training from scratch and using embeddings pretrained on 840 billion words, and this improvement is consistent on both datasets. Compared to the SQuAD results, an important difference is that spelling was not as useful on SNLI and MultiNLI. We also note that we tried using fixed random embeddings for OOV words as proposed by (Dhingra et al., 2017), and that this method did not bring a significant advantage over the baseline.

In order to gain some insights on the performance of our entailment recognition models, in Figure 3a we plot a t-SNE (van der Maaten & Hinton, 2008) visualization of word embeddings computed for the words from BLESS dataset (Baroni & Lenci, 2011). Specifically, we used embeddings produced

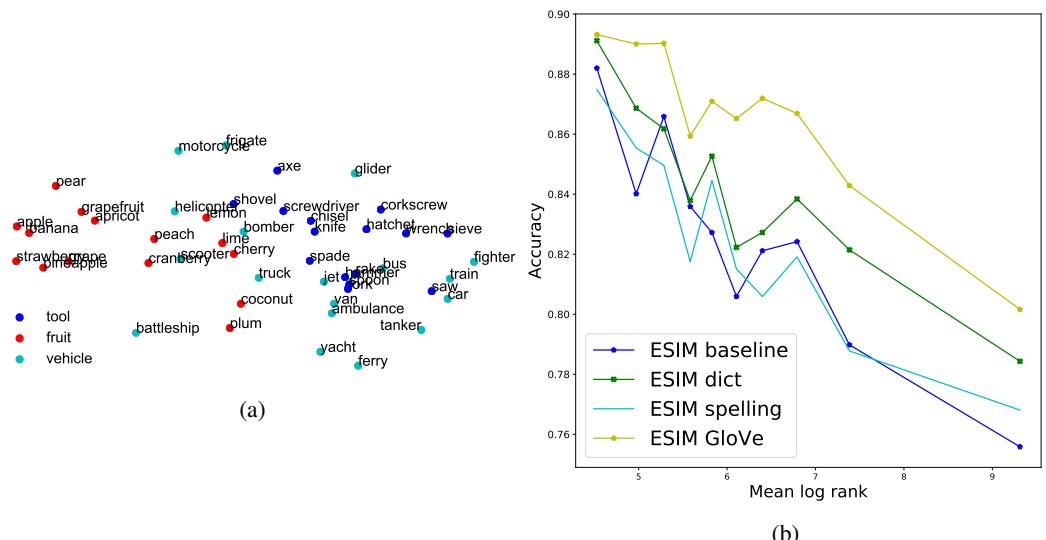

Figure 3: (a) t-SNE projection of word embeddings computed on the fly. (b) Prediction accuracy of different ESIM models trained on SNLI, as a function of the mean log rank of words in the input. As expected, the dictionary-enabled model clearly outperforms the baseline on sentences contaning rare words.

by the definition encoder of our best SNLI model using Wordnet definitions. The BLESS dataset contains three categories of words: fruits, tools and vehicles. One can see that fruit words tend to be separated from tool and vehicle words. Figure 3b shows that, as expected, dictionary-enabled models significantly outperform baseline models for sentences containing rare words.

## 4.3 LANGUAGE MODELLING

Experiments in the previous sections used datasets of moderate size. To get an idea of how useful different auxiliary data sources will be for datasets of various sizes, we also apply our approach to the One Billion Words (OBW) language modelling task (Chelba et al., 2014). In the first round of experiments, we use only 1% of the training data ($\sim 10^7$ words) and in the second we train our models on the whole training set ($\sim 10^9$ words). Similarly to prior work on using the spelling (Ling et al., 2015) we restrict the softmax output layer to only predict probabilities of the 10k most common words, however, we do not impose such a constraint when the model processes the words from the input context. We want to assess thereby if having a definition of an observed word helps the model to predict the following ones from a restricted vocabulary.

Our baseline model is an LSTM with 500 units and with trainable input embeddings for the 10k most frequent input words. This covers around 90.24% of all word occurrences. We consider computing embeddings of the less frequent input words from their dictionary definitions, GloVe vectors and spellings. These sources of auxiliary information were available for 63.35%, 97.43% and 100% of the rest of occurrences respectively. In order to compare how helpful these sources are when they are available, we run additional set of experiments with "restricted" inputs. Specifically, we only use auxiliary information for a word if it has both a GloVe embedding and a dictionary definition. When the word does not have any of these, we replace it with "UNK". We report results for three variants of dictionary-enabled models. The first variant (dict1) uses the same LSTM for reading the text and the definitions. The second one (dict2) has two separate LSTMs but the word embeddings are shared[6]. The third variant (dict+spelling) adds spelling to our best dictionary model. Lastly, we trained a model that used the lowercased lemma of the word as the definition.

The training was early-stopped using a development set. We report the test perplexities in Table 3. Similarly to our other experiments, using external information to compute embeddings of unknown

---

[6]We also tried learning separate embeddings in both cases, as well as building a separate vocabulary but this did not help.

Table 3: Results on 1% of OBW and full OBW training set. We report 2 variants of perplexities on the test set. The experiments where coverage is restricted are indicated by (R). See Section 4.3 for explanations.

| model | 1% | | 100% | |
|---|---|---|---|---|
| | PPL | PPL after OOV | PPL | PPL after OOV |
| baseline | 73.07 | 102.09 | 46.77 | 70.76 |
| lemma+lowercase | 70.57 | 86.24 | 45.02 | 56.45 |
| spelling | 67.06 | 55.04 | 39.77 | 26.27 |
| glove | 63.83 | 44.26 | 39.15 | 24.20 |
| dict1 | 69.14 | 68.38 | 42.04 | 36.50 |
| dict2 | 68.08 | 65.77 | 41.34 | 34.47 |
| dict2+spelling | 66.23 | 52.06 | 39.56 | 25.22 |
| spelling (R) | 68.24 | 63.65 | 41.34 | 33.61 |
| glove (R) | 66.69 | 57.21 | 41.22 | 32.35 |
| dict2 (R) | 68.06 | 65.62 | 41.34 | 34.53 |
| dict2+spelling (R) | 67.43 | 60.58 | 40.89 | 32.48 |

words helps in all cases. We observe a significant gain even for dict1, which is remarkable as this model has the same architecture and parameters as the baseline. We note that lemma+lowercase performs worse than any model with the dictionary, which suggests that dictionary definitions are used in a non-trivial way. Adding spelling consistently helps more than adding dictionary definitions. In our experiments with restricted inputs ((R) in Table 3), spelling and dict2 show similar performance, which suggests that this difference is mostly due to the complete coverage of spelling. Using both dictionary and spelling is consistently slightly better than using just spelling, and the improvement is more pronounced in the restricted setting. Using GloVe embeddings results in the best perplexity. Switching to the full training set shrank all the gaps.

To zoom in on how the models deal with rare words, we look at the perplexities of the words that appear right after out-of-vocabulary words (PPL after OOV). We can see that the ranking of different models mostly stays the same, yet the differences in performance become larger. For example, on the 1% version of the dataset, in the restricted setting, adding definitions to the spelling helps to bridge half of the 6 points PPL after OOV gap between spelling and GloVe. This is in line with our expectations that when definitions are available they should be helpful for handling rare words.

## 5 DISCUSSION

We showed how different sources of auxiliary information, such as the spelling and a dictionary of definitions can be used to produce on the fly useful embeddings for rare words. While it was known before that adding the spelling information to the model is helpful, it is often hard or not possible to infer the meaning directly from the characters, as confirmed by our entailment recognition experiments. Our more general approach offers endless possibilities of adding other data sources and learning end-to-end to extract the relevant bits of information from them. Our experiments with a dictionary of definitions show the feasibility of the approach, as we report improvements over using just the spelling on question answering and semantic entailment classification tasks. Our qualitative investigations on the question answering data confirms our intuition on where the improvement comes from. It is also clear from them that adding more auxiliary data would help, and that it would probably be also useful to add definitions not just for words, but also for phrases (see "Mark Twain" from Section 4.1). We are planning to add more data sources (e.g. first sentences from Wikipedia articles) and better use the available ones (WordNet has definitions of phrasal verbs like "come across") in our future work.

An important question that we did not touch in this paper is how to deal with rare words in the auxiliary information, such as dictionary definitions. Based on our qualitative investigations (see the example with "arrow" and "weapon" in Section 4.1), we believe that better handling rare words in the auxiliary information could substantially improve the proposed method. It would be natural to

use on the fly embeddings similarly to the ones that we produce for words from the input, but the straight-forward approach of computing them on request would be very computation and memory hungry. One would furthermore have to resolve cyclical dependencies, which are unfortunately common in dictionary data (when e.g. "entertainment" is defined using "diverting" and "diverting" is defined using "entertainment"). In our future work we want to investigate asynchronous training of on the fly embeddings and the main model.

## 6 CONCLUSION

In this paper, we have shown that introducing relatively small amounts of auxiliary data and a method for computing embeddings on the fly using that data bridges the gap between data-poor setups, where embeddings need to be learned directly from the end task, and data-rich setups, where embeddings can be pretrained and sufficient external data exists to ensure in-domain lexical coverage. A large representative corpus to pretrain word embeddings is not always available and our method is applicable when one has access only to limited auxiliary data. Learning end-to-end from auxiliary sources can be extremely data efficient when these sources represent compressed relevant information about the word, as dictionary definitions do. A related desirable aspect of our approach is that it may partially return the control over what a language processing system does into the hands of engineers or even users: when dissatisfied with the output, they may edit or add auxiliary information to the system to make it perform as desired. Furthermore, domain adaptation with our method could be carried out simply by using other sources of auxiliary knowledge, for example definitions of domain-specific technical terms in order to understand medical texts. Overall, the aforementioned properties of our method make it a promising alternative to the existing approaches to handling rare words.

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
