# OpenReview forum: "Learning to Compute Word Embeddings On the Fly"
_ICLR.cc/2018/Conference — Reject_

### Official Review · AnonReviewer3 · 2017-11-27

**Rating:** 5
**Confidence:** 4

**Review:**

This paper describes a method for computing representations for out-of-vocabulary words, e.g. based on their spelling or dictionary definitions. The main difference from previous approaches is that the model is that the embeddings are trained end-to-end for a specific task, rather than trying to produce generically useful embeddings. The method leads to better performance than using no external resources, but not as high performance as using Glove embeddings. The paper is clearly written, and has useful ablation experiments. However, I have a couple of questions/concerns:
- Most of the gains seem to come from using the spelling of the word. As the authors note, this kind of character level modelling has been used in many previous works.
- I would be slightly surprised if no previous work has used external resources for training word representations using an end-task loss, but I don’t know the area well enough to make specific suggestions
- I’m a little skeptical about how often this method would really be useful in practice. It seems to assume that you don’t have much unlabelled text (or you’d use Glove), but you probably need a large labelled dataset to learn how to read dictionary definitions well. All the experiments use large tasks - it would be helpful to have an experiment showing an improvement over character-level modelling on a smaller task.
- The results on SQUAD seem pretty weak - 52-64%, compared to the SOTA of 81. It seems like the proposed method is quite generic, so why not apply it to a stronger baseline?

---

### Official Review · AnonReviewer1 · 2017-11-27
**Good practical approach for rare (and unseen) word handling**

**Rating:** 7
**Confidence:** 3

**Review:**


This paper illustrates a method to compute produce word embeddings on the fly for rare words, using a pragmatic combination of existing ideas:

* Backing off to a separate decoder for rare words a la Luong and Manning (https://arxiv.org/pdf/1604.00788.pdf, should be cited, though the idea might be older).

* Using character-level models a la Ling et al.

* Using dictionary embeddings a la Hill et al.

None of these ideas are new before but I haven’t seen them combined in this way before. This is a very practical idea, well-explained with a thorough set of experiments across three different tasks. The paper is not surprising but this seems like an effective technique for people who want to build effective systems with whatever data they’ve got.

---

### Official Review · AnonReviewer2 · 2017-11-28
**Nice examination of learning on-the-fly word embeddings, but a fairly small focused contribution**

**Rating:** 5
**Confidence:** 4

**Review:**

This paper examines ways of producing word embeddings for rare words on demand. The key real-world use case is for domain specific terms, but here the techniques are demonstrated on rarer words in standard data sets. The strength of this paper is that it both gives a more systematic framework for and builds on existing ideas (character-based models, using dictionary definitions) to implement them as part of a model trained on the end task.

The contribution is clear but not huge. In general, for the scope of the paper, it seems like what is here could fairly easily have been made into a short paper for other conferences that have that category. The basic method easily fits within 3 pages, and while the presentation of the experiments would need to be much briefer, this seems quite possible. More things could have been considered. Some appear in the paper, and there are some fairly natural other ones such as mining some use contexts of a word (such as just from Google snippets) rather than only using textual definitions from wordnet. The contributions are showing that existing work using character-level models and definitions can be improved by optimizing representation learning in the context of the final task, and the idea of adding a learned linear transformation matrix inside the mean pooling model (p.3). However, it is not made very clear why this matrix is needed or what the qualitative effect of its addition is.

The paper is clearly written.

A paper that should be referred to is the (short) paper of Dhingra et al. (2017): A Comparative Study of Word Embeddings
for Reading Comprehension https://arxiv.org/pdf/1703.00993.pdf . While it in no way covers the same ground as this paper it is relevant as follows: This paper assumes a baseline that is also described in that paper of using a fixed vocab and mapping other words to UNK. However, they point out that at least for matching tasks like QA and NLI that one can do better by assigning random vectors on the fly to unknown words. That method could also be considered as a possible approach to compare against here.

Other comments:
 - The paper suggests a couple of times including at the end of the 2nd Intro paragraph that you can't really expect spelling models to perform well in representing the semantics of arbitrary words (which are not morphological derivations, etc.). While this argument has intuitive appeal, it seems to fly in the face of the fact that actually spelling models, including in this paper, seem to do surprisingly well at learning such arbitrary semantics.
 - p.2: You use pretrained GloVe vectors that you do not update. My impression is that people have had mixed results, sometimes better, sometimes worse with updating pretrained vectors or not. Did you try it both ways?
 - fn. 1: Perhaps slightly exaggerates the point being made, since people usually also get good results with the GloVe or word2vec model trained on "only" 6 billion words – 2 orders of magnitude less data.
 - p.4. When no definition is available, is making e_d(w) a zero vector worse than or about the same as using a trained UNK vector?
 - Table 1: The baseline seems reasonable (near enough to the quality of the original Salesforce model from 2016 (66 F1) but well below current best single models of around 76-78 F1. The difference between D1 and D3 does well illustrate that better definition learning is done with backprop from end objective. This model shows the rather strong performance of spelling models – at least on this task – which he again benefit from training in the context of the end objective.
 - Fig 2: It's weird that only the +dict (left) model learns to connect "In" and "where". The point made in the text between "Where" and "overseas" is perfectly reasonable, but it is a mystery why the base model on the right doesn't learn to associate the common words "where" and "in" both commonly expressing a location.
 - Table 2: These results are interestingly different. Dict is much more useful than spelling here. I guess that is because of the nature of NLI, but it isn't 100% clear why NLI benefits so much more than QA from definitional knowledge.
 - p.7: I was slightly surprised by how small vocabs (3k and 5k words) are said to be optimal for NLI (and similar remarks hold for SQuAD). My impression is that most papers on NLI use much larger vocabs, no?
 - Fig 3: This could really be drawn considerably better: make the dots bigger and their colors more distinct.
 - Table 3: The differences here are quite small and perhaps the least compelling, but the same trends hold.

---

### Author Response · Authors · 2017-12-28
**Rebuttal**

We are grateful to the reviewers for their thorough and thoughtful reviews! Based on their feedback, we uploaded a revised version of the paper with a number of small changes.
In the rest of the rebuttal we address some of the concerns that reviewers raised. We conclude by restating the strengths of the paper.

AnonReviewer 2 (R2) asked what the contribution of training a linear transformation of mean pooling is. Our understanding is that such a linear transformation helps to compensate for the difference between the trainable word embeddings and their complements that are obtained by averaging embeddings of the words from definitions.

We thank R2 for pointing at the paper by Dhingra et al, which proposes to use fixed random embeddings for OOV words. We updated the paper to include this reference. As mentioned by R2, this technique does not really cover the same ground, because it only allows to match exactly identical words against each other, whereas our method takes into account semantics of the OOV words, as expressed in the definitions. As suggested by R2, we carried out an additional experiment on SNLI to verify our reasoning, and we did not find fixed random embeddings helpful. This additional experiment has been mentioned in the paper.

R2 also commented on the fact that in some of our experiments spelling is more helpful than in others. They also suggested that this contradicts our initial argument that semantics can not always be inferred from spelling. We respectfully disagree for the following reasons: (a) the fact that gains from using the spelling and the definition are complementary is aligned  with our expectation that spelling is not sufficient, (b) how much different sources of auxiliary information help is highly dataset-specific, and high performance of spelling on SQuAD can be due to the fact that a lot of questions can be answered by looking for question words in the document (Weissenborn et al, 2017), not because semantics could be inferred from spelling, (c) our qualitative investigation shows that the dictionary does enable semantic processing that spelling does not permit, such as matching “where” and “overseas”, (d) NLI, arguably the most semantically demanding of the considered benchmarks, shows a clear superiority of a dictionary-enabled model. We also make similar arguments in the paper, for example in Sentence 2 of Section 5.

We thank R2 for asking about vocabulary sizes; indeed, we used a 20k vocabulary for MultiNLI, which fact is reflected in the new edition of the paper. We did however find that using more than 3k (5k) on SQuAD (SNLI) merely caused stronger overfitting. Vocabulary sizes may be larger in other papers due to the fact they rely on embeddings that were pretrained on huge corpora.

We thank AnonReviewer1 for their positive review of the paper. We note that we do not study character-level decoders in our work, focusing only on the model’s ability to understand OOV inputs. In the light of this, we are not sure that the work of Luong and Manning is a required citation. We do cite Ling et al as a prior work on computing word representations from characters.

We do not fully agree with AnonReviewer3 (R3) when they say “most of the gains seem to come from using the spelling of the word”.  As mentioned above in this rebuttal, in our NLI experiments dictionary definitions were a lot more helpful than spelling, and besides, in other experiments it was shown that benefits from the spelling and dictionary definitions are complementary. With regard of the size of the datasets that we used, our language modelling experiments suggest that the proposed technique is only more helpful for small datasets. Lastly, while we completely agree that it would be interesting to apply the proposed method to SOTA models for SQuAD, we note that SOTA on this dataset has been improving rapidly over the last year. It’s challenging to keep up with SOTA in an on-going research project, and besides our approach is by no means model specific, which makes us expect that the reported results should transfer across all SQuAD models.

To conclude, we would like to reiterate our key arguments in favor of the acceptance of this work. The paper proposes a conceptually simple, yet novel method to tackle a very general problem of OOV words in natural language processing. The experimental results that we provide on QA, NLI and language modelling give the reader an idea of whether this method is applicable to their domain of interest. Under a reasonable assumption that NLI recognition was the most semantically demanding task out of the considered ones, the relevance of the proposed method will only grow as the progress in the field will allow using harder datasets and tasks. Lastly, we believe that our method will be especially helpful for practitioners working in technical domains, such as legal text and biological texts, where exact definitions should typically be available.

---

### Decision · Program_Chairs · 2018-01-29
**ICLR 2018 Conference Acceptance Decision**

**Decision:**

Reject

**Comment:**

The pros and cons of the paper can be summarized as follows:

Pros:
* The method of combining together multiple information sources is effective
* Experimental evaluation is thorough

Cons:
* The method is a relatively minor contribution, combining together multiple existing methods to improve word embeddings. This also necessitates the model being at least as complicated as all the constituent models, which might be a barrier to practical applicability

As an auxiliary comment, the title and emphasis on computing embeddings "on the fly" is a bit puzzling. This is certainly not the first paper that is able to calculate word embeddings for unknown words (e.g. all the cited work on character-based or dictionary-based methods can do so as well). If the emphasis is calculating word embeddings just-in-time instead of ahead-of-time, then I would also expect an evaluation of the speed or memory requirements benefits of doing so. Perhaps a better title for the paper would be "integrating multiple information sources in training of word embeddings", or perhaps a more sexy paraphrase of the same.

Overall, the method seems to be solid, but the paper was pushed out by other submissions.